# Biosynthesis of Amino Acids in *Xanthomonas oryzae* pv. *oryzae* Is Essential to Its Pathogenicity

**DOI:** 10.3390/microorganisms7120693

**Published:** 2019-12-13

**Authors:** Ting Li, Zhaohong Zhan, Yunuan Lin, Maojuan Lin, Qingbiao Xie, Yinhua Chen, Chaozu He, Jun Tao, Chunxia Li

**Affiliations:** 1Hainan Key Laboratory for Sustainable Utilization of Tropical Bioresources, Haikou 570228, Chinayhchen@hainanu.edu.cn (Y.C.);; 2College of Tropical Crops, Hainan University, Haikou 570228, China

**Keywords:** *Xanthomonas oryzae*, amino acid synthesis, *Xoo*-rice interaction, leucine

## Abstract

*Xanthomonas oryzae* pv. *oryzae* (*Xoo*) is the causal agent of rice bacterial blight disease, which causes a large reduction in rice production. The successful interaction of pathogens and plants requires a particular nutrient environment that allows pathogen growth and the initiation of both pathogen and host responses. Amino acid synthesis is essential for bacterial growth when bacteria encounter amino acid-deficient environments, but the effects of amino acid synthesis on *Xoo* pathogenicity are unclear. Here, we systemically deleted the essential genes (*leuB*, *leuC*, *leuD*, *ilvC*, *thrC*, *hisD*, *trpC*, *argH*, *metB*, and *aspC*) involved in the synthesis of different amino acids and analyzed the effects of these mutations on *Xoo* virulence. Our results showed that leucine, isoleucine, valine, histidine, threonine, arginine, tryptophan, and cysteine syntheses are essential to *Xoo* infection. We further studied the role of leucine in the interaction between pathogens and hosts and found that leucine could stimulate some virulence-related responses and regulate *Xoo* pathogenicity. Our findings highlight that amino acids not only act as nutrients for bacterial growth but also play essential roles in the *Xoo* and rice interaction.

## 1. Introduction

Amino acids play important roles in cellular metabolism and stress responses. Some plants subjected to stress accumulate proline and other amino acids that act as osmolytes, regulate ion transport, modulate stomata opening, and detoxify heavy metals [1]. Amino acids also affect the synthesis and activities of some enzymes, gene expression, and redox homeostasis [1]. In addition, some amino acids in plants, including leucine and glutamate, may contribute to feedback inhibition by pathogens [2,3]. Exogenous proteinogenic amino acids can induce systemic resistance in rice [4], indicating that amino acids are key factors for plant resistance to pathogens.

In *Mycobacterium tuberculosis*, mutation of *metA*, which encodes homoserine transacetylase, an enzyme that is involved in methionine and S-adenosylmethionine biosynthesis, renders this pathogen sensitive to killing in immunocompetent or immunocompromised mice, leading to its rapid clearance from host tissues [5]. Similarly, arginine biosynthesis pathway mutants (Δ*argB* and Δ*argF*) are also unable to scavenge host arginine and are quickly cleared from both immunocompetent and immunocompromised mice [6,7]. The lysine-auxotroph strain (Δ*lysF*) of *Aspergillus fumigatus* or *A. nidulans* is unable to grow in the host [8]. Leucine, a gamma-aminobutyric acid (GABA), not only regulates intracellular signaling pathways but also has important effects on protein synthesis and immunity in plants [3]. Disruption of leucine synthesis influences iron homeostasis and thus attenuates the virulence of the fungus *Cryptococcus neoformans* [9]. Another report has also shown that leucine synthesis is essential to the growth of *Acidovorax citrulli* in planta [10]. These findings suggest that amino acid synthesis is important for bacteria and fungi to successfully infect hosts.

*Xanthomonas oryzae* pv. *oryzae* (*Xoo*) is a Gram-negative bacterium and the causal agent of rice bacterial blight disease, which causes a large reduction in rice production around the world. Plant diseases are caused by the interaction of pathogens and plants in a particular nutrient environment, which is dependent on the interaction between plant and pathogen factors. Although a previous study has shown that histidine can affect the pathogenicity of *Xanthomonas oryzae* pv. *oryzicola* (*Xoc*) [11], the roles of amino acids in the *Xoo*–rice interaction remain unclear.

In this study, we identified and functionally characterized the essential genes for *Xoo* amino acid synthesis. We found that synthesis of leucine, isoleucine, valine, histidine, threonine, arginine, tryptophan, and cysteine is important for *Xoo* infection. We also analyzed the roles of branched-chain amino acids (BCAAs), especially leucine, in the *Xoo*–rice interaction in detail and found that leucine can regulate diverse pathways to influence *Xoo* virulence. Our data indicate that amino acids are not only essential nutrients for *Xoo*, but also regulate virulence-related pathways to influence its infection processes. Our study establishes a solid link between primary metabolism and bacterial pathogenicity and provides new ideas for protecting rice from *Xoo* infection.

## 2. Materials and Methods

### 2.1. Bacterial Culture and Growth Conditions

The plasmids and bacterial strains used in this study are listed in Appendix A. *Xoo* was cultured on peptone sucrose agar (PSA) or M4N minimal medium [12,13] at 28 °C with the appropriate antibiotic. *Escherichia coli* was cultured at 37 °C and 200 rpm in lysogenic broth (LB). Kanamycin (50 mg/L) and spectinomycin (100 mg/L) were added when appropriate.

### 2.2. Xoo Mutation and Complementation

*Xoo* deletion mutants were generated in PXO99A by marker-free exchange mutagenesis using the suicide vector pK18*mobSacB* [12,14]. Approximately 500-bp sequences flanking both sides of the region to be deleted were amplified by PCR (all of the primers are presented in Appendix A) and cloned into pK18mobSacB using the restriction enzyme sites *Hind*III/*BamH*I and *BamH*I/*EcoR*I. *Xoo* cells were transformed with suicide plasmids by electroporation and plated on PGA (peptone glucose-agar) containing kanamycin (20 mg/L) [12]. After two rounds of recombination, the open reading frame was deleted from genomic DNA. Mutants were confirmed by PCR and DNA sequencing.

For construction of the plasmid for complementation of the *leuB* mutant, whole length fragments of *PXO_02610* (*leuB*) were amplified by PCR and cloned into a broad host-range vector pHM1 [15].

### 2.3. Virulence Assays

*Xoo* pathogenicity was determined as previously described [12]. Briefly, 60-day-old leaves from the rice cultivar IR24 were inoculated using the leaf clipping method. Bacteria for inoculation were taken from PSA plates and resuspended in sterile water at an optical density at 600 nm (OD_600_) of 0.6. Twenty expanded upper leaves of rice were inoculated. Water-soaked lesions were measured at 14 days post-inoculation. Three biological replicates were measured for each strain.

### 2.4. Raw Amino Acid Extraction from Rice Leaves and Bacterial Growth Assays

Fresh rice leaves (~0.3 g) were fully cut, thoroughly ground, resuspended in 15 mL sterile water, and sterilized by 0.22 μM filtration to obtain raw extracts of amino acids. The concentrations of the various amino acids in rice leaves were detected by liquid chromatography-tandem mass spectrometry or an amino acid analyzer [16] carried out by Qingdao Sci-Tech Innovation Quality Inspection Co., Ltd. (Qingdao, China). The extracts were added to amino acid-free solid M4N medium at a ratio of 1:4 of the extracts to media to analyze their effects on *Xoo* growth. Bacterial cultures were spotted and cultured for three days at 28 °C.

### 2.5. RNA Sequencing (RNA-Seq) and Real-Time Quantitative RT-PCR

The Δ*leuB* strains were first grown in PSA to OD_600_~0.8, washed twice with M4N media, and regrown for 30 min in M4N medium supplemented with 250 µM leucine or not. Total RNA was extracted using an RNA purification kit (QIAGEN, Shanghai, China) according to the manufacturer’s protocol. cDNA was generated from 5 μg of RNA using SuperScript III reverse transcriptase (Invitrogen, Carlsbad, CA, USA). RNA-Seq was carried out by Guangzhou RiboBio Co., LTD (Guangzhou, China). Transcript quantification was performed by real-time quantitative RT-PCR using Maxima SYBR Green/ROX qPCR Master Mix in an ABI 7500 sequence detection system (Thermo Fisher Scientific, Shanghai, China). The results were normalized against the *rpoD* gene.

### 2.6. GUS Activity Assays

The *motA*, *cheL*, and *ilvC* promoter region containing the translation start site was amplified by PCR and ligated into the promoter probe plasmid pK18GUS [12]. The plasmids were transformed into PXO99A and Δ*leuB*. The resulting strains were grown in M4N medium supplemented with 250 µM leucine or not. Cells were collected by centrifugation, and GUS activity was assayed as previously described [12,17].

### 2.7. Growth Assays

Seven streaked strains of wild-type PXO99A and the mutants Δ*PX2610*, Δ*PX0822*, Δ*PX0833*, Δ*PX4004*, Δ*PX0346*, Δ*PX3157* were picked from plates and inoculated into PSA medium and were then grown for 36 h with shaking at 28 °C. Bacterial cultures were spotted and cultured for three days at 28 °C on M4N agar plates, which were supplemented with amino acids with respective concentrations of 0, 50, 150, and 250 µM. The growth rates were measured by taking aliquots of the cultures at approximately 6-h intervals until 60 h, and the OD_600_ values of each strain were recorded and plotted versus time.

## 3. Results

### 3.1. Identification of Essential Genes for Amino Acid Synthesis in Xoo

To analyze the effects of amino acids on *Xoo* virulence, we first constructed strains that cannot synthesize one or more amino acids. Using the amino acid biosynthesis pathways of *Xoo* from the KEGG website (https://www.kegg.jp/kegg-bin/show_pathway?xop01230), we chose genes that are predicted to encode key enzymes for amino acid syntheses to mutate (Table 1). Because the key genes that control the synthesis of some amino acids were not well predicated in *Xoo*, we did not choose all amino acids as study targets. We first focused on leucine, threonine, histidine, tryptophan, arginine, and cysteine and analyzed the essential genes required for the biosynthesis of these amino acids (Table 1). We constructed deletion mutants of the corresponding genes (*leuB*, *leuCD*, *thrC*, *hisD*, *trpC*, *argH*, and *metB*; Appendix A) and investigated whether these genes were required for synthesis of the corresponding amino acids. Finally, we tested the growth of the mutants in minimal medium (M4N) with different concentrations of the corresponding amino acid. Compared with wild-type PXO99A, which grew well in M4N medium without leucine, threonine, histidine, tryptophan, arginine, and cysteine, Δ*PX2610* (Δ*leuB*), Δ*PX0822* (Δ*thrC*), Δ*PX0833* (Δ*hisD*), Δ*PX4004* (Δ*trpC*), Δ*PX0346* (Δ*argH*), Δ*PX3157* (Δ*metB*) showed significant growth inhibition under the corresponding amino acid deficiency conditions (Figure 1). Five of the mutants (Δ*leuB*, Δ*thrC*, Δ*hisD*, Δ*trpC*, and Δ*argH*) could not grow in M4N medium without supplementation with leucine, threonine, histidine, tryptophan, or arginine, respectively, after three days of incubation, suggesting that these genes are essential for synthesis of the corresponding amino acids. One mutant, Δ*metB*, showed significantly reduced growth without L-cysteine, indicating that other genes and pathways might also be involved in cysteine synthesis. The growth of the Δ*hisD*, Δ*trpC*, and Δ*metB* mutants could be restored by supplementation with 50 µM of histidine, tryptophan, and cysteine, respectively. While Δ*leuB*, Δ*thrC*, and Δ*argH* did not grow well in M4N supplemented with the same concentration of leucine, threonine, and arginine, respectively, indicating that the transportation rates or usage of these amino acids may be different in *Xoo* (Figure 1). As a result, our findings highlight that *leuB*, *thrC*, *hisD*, *trpC*, *argH*, and *metB* are involved in the biosynthesis of leucine, threonine, histidine, tryptophan, arginine, and cysteine, respectively, in *Xoo*.

### 3.2. Relationship between Amino Acids Synthesis and the Pathogenicity of Xoo

To determine whether synthesis of amino acids plays an important role in *Xoo* virulence, we compared the virulence of the corresponding auxotrophic strains with the wild-type strain. As shown in Figure 2, all the test mutants exhibited different degrees of decreased virulence in comparison with wild-type PXO99A, as measured by the length of the water-soaking lesion. Specifically, Δ*leuB* and Δ*leuCD* almost lost their infection abilities, and Δ*thrC*, Δ*argH*, and Δ*ilvC* showed significant growth inhibition compared with PXO99A. These data demonstrate that synthesis of amino acids is important for *Xoo* infection. To further confirm that leucine synthesis is essential to *Xoo* virulence, we constructed the Δ*leuB* complemented strain C-Δ*leuB* and found that this strain retained almost a full infection ability (Figure 2B,D), suggesting that the lack of leucine synthesis caused by the *leuB* mutation resulted in *Xoo* growth defects in planta. No virulence difference was found among Δ*lrp*, Δ*aspC*, and PXO99A, indicating that Lrp, which is predicted to be a leucine-response protein, is not a virulence-regulated regulator and that AspC, which may be involved in different amino acid metabolism pathways, is also not a contributor to *Xoo* pathogenicity.

### 3.3. Free Amino Acids in Rice Leaves Tightly Influence Xoo Growth

The reduced virulence of the amino acid auxotrophic strains indicates that rice leaves cannot provide a sufficient amount of the corresponding amino acids. To confirm this finding, we first measured the concentration of various amino acids in rice leaves (Figure 3A). Except for increased arginine (~1.1 g/kg) and cysteine (~0.5 g/kg) contents, the contents of the remaining amino acid in rice leaves were less than 0.2 g/kg, and methionine, threonine, histidine, isoleucine, and valine were even lower than 0.1 g/kg. Then, we detected the growth states of the *Xoo* strains in M4N media supplemented with raw extracts of amino acids from rice leaves. Wild-type PXO99A and the deletion mutants grew well in M4N media supplemented with 250 µM amino acids. Significant growth inhibition was observed in Δ*leuB*, Δ*leuCD*, Δ*thrC*, Δ*hisD*, Δ*trpC*, and Δ*ilvC* when the raw extracts of amino acids from rice leaves were used as the sole nitrogen source (Figure 3B). This result may be due to the low concentrations of leucine, threonine, histidine, tryptophan, isoleucine, and valine in rice and/or the insufficient transportation of these nutrients by these mutants. Therefore, our results illustrate that limiting amino acid availability in rice leaves may restrict *Xoo* growth *in planta*.

### 3.4. Xoo Has Different Abilities to Utilize Various Amino Acids

As the virulence phenotypes of some amino acid auxotrophic strains are not explained by the content of amino acids in rice leaves, we hypothesize that *Xoo* cannot efficiently utilize these amino acids. Thus, we tested how much amino acids in the environment could afford the growth of these *Xoo* strains. The growth rates of PXO99A and the mutants (Δ*thrC*, Δ*hisD*, Δ*trpC*, Δ*argH*, and Δ*metB*) in M4N supplemented with different concentrations of leucine, threonine, histidine, tryptophan, arginine, and cysteine are shown in Figure 4. Almost all the bacteria grew faster as the concentration of amino acids increased. Notably, Δ*leuB* and Δ*thrC* did not grow well in the presence of 50 µM leucine and threonine, respectively. Moreover, these mutants also grew slower than the wild-type strain even in the presence of 250 µM of the corresponding amino acids. These results suggest that the reduced growth rates of these auxotrophy strains, even at high amino acid concentrations, may be the reason that these strains had a worse inflectional ability compared with that of the wild-type strain.

### 3.5. Leucine Regulates Virulence-Related Pathways

As some amino acids can function as nutrients as well as regulatory molecules in higher organisms [18,19], we determined whether they could also regulate pathways involved in *Xoo* virulence. Because the regulatory role of leucine has been extensively studied in higher plants [9,10] and *Xoo* Δ*leuB* has no virulence in rice (Figure 2), we chose this mutant for the subsequent experiments. Although Lrp has been confirmed to be a leucine-response regulator in different bacteria [20,21,22], the *lrp* mutant does not affect *Xoo* virulence or growth (Figure 2), suggesting that Lrp-dependent leucine signaling might not be important for *Xoo* infection and that other pathways might be involved in leucine-dependent signaling. Therefore, we first performed RNA sequencing (RNA-Seq) analysis to identify which genes were deferentially expressed in Δ*leuB* after induction of leucine (Appendix A). Then, we used real-time PCR to further detect these changes (Figure 5). Δ*leuB* was grown in M4N medium supplemented with or without 250 µM leucine for 30 min. When no leucine was in the medium, the expression of two type VI secretion system-related genes, *PXO_00256* (*vgrG*) and *PXO_00264* (*hcp*), and two amino acid synthesis genes, *PXO_00835* (*hisB*) and *PXO_01963* (*gluL*), were significantly inhibited, indicating that the type VI secretion system and the synthesis of some amino acids (especially histidine and glutamine) are positively regulated by leucine. These data may explain why histidine synthesis plays an essential role in *Xoc* infection [11]. In contrast, the expression of two genes (*PXO_00728*, *hutG*, and *PXO_03319*, *gltB*) involved in glutamate synthesis was inhibited in the absence of leucine. The mRNA levels of *gluL*, whose encoding protein converts glutamate into glutamine, were elevated under the same conditions, indicating that leucine might regulate the ratio of glutamate to glutamine in *Xoo*. In addition, chemotaxis- and motility-related genes, including *PXO_00029* (*cheW*), *PXO_00055* (*cheR*), *PXO_01007* (*flgG*), and *PXO_06210* (*cheY*) were inhibited at the transcriptional level in the absence of leucine, implying that leucine can influence *Xoo* chemotaxis and thus its pathogenicity. As expected, the expression of genes (*PXO_02605*, *ilvC*, and *PXO_03941*, *ilvD*) involved in BCAA synthesis was stimulated by a lack of leucine because of feedback effects. To further confirm these results, we used β-glucuronidase (GUS) activity assays to analyze the corresponding promoter activities in the absence of leucine (Figure 5B and Appendix A). We chose the promoters of the chemotaxis operons *motA* and *cheL* and the BCAA synthesis operon (*ilvC*) and found that all the expression levels were increased in the absence of leucine (Figure 5B), in line with the qRT-PCR results. Taken together, our data suggest that leucine can regulate the expression of genes involved in chemotaxis and type VI secretion and crosstalk with the biosynthesis pathways of other amino acids (Appendix A).

## 4. Discussion

### 4.1. Relationship between Amino Acid Synthesis and Xoo Virulence

Although many bacteria have the ability to synthesize amino acids, environmentally free amino acids can first be consumed by these organisms. Free available amino acids are not only nutrients, but also regulate diverse pathways in different organisms [1,3,18,19,20,21,22]. The effective infection of *Xoo* in rice depends on its successful growth in planta and requires the activation of some virulence-related processes [11,12,13]. Mutations of genes involved in the biosynthesis pathways of several amino acids decreased *Xoo* virulence (Figure 2), which implies that free amino acids in rice leaves may not be sufficient for *Xoo* growth in planta. This result was confirmed when we used raw amino acid extracts from rice leaves to cultivate the corresponding auxotrophic strains (Figure 3). However, the amino acids in rice leaves are not always consistent with the pathogenicity of auxotrophic strains (Figure 2 and Figure 3). For example, the levels of arginine and cysteine in rice leaves are far greater than those of histidine and tryptophan, but *argH* and *metB* mutants have a lower infectional ability than that of *hisD* and *trpC* mutants (Figure 3). One possible reason for this finding is that *Xoo* has different transport or usage efficiencies of different amino acids. When we cultivated these mutants in different concentrations of the corresponding amino acids, the *argH* mutant grew slower than the *hisD* and *trpC* mutants. However, the *metB* mutant grew faster than the other tested strains, including the wild-type strain, in the presence of the same concentration of the corresponding amino acids (Figure 4). These data suggest that the roles of amino acids in *Xoo* virulence are not only nutrient suppliers but also regulatory molecules. Moreover, the data shown in Figure 5 indicate that leucine can regulate diverse pathways, as found in other organisms [10,19,20,21], suggesting that leucine, as well as other amino acids, may be an important factor in regulating the primary metabolism and pathogenicity of *Xoo*.

### 4.2. Pathways for BCAA Synthesis in Xoo

As Δ*leuB* lost virulence and did not grow in the absence of leucine (Figure 3 and Figure 4), we believe that the synthesis of other forms of BCAA is also essential to *Xoo* infection. The ketol-acid reductoisomerase PXO_02605 (IlvC) is the putative key enzyme involved in valine, leucine, and isoleucine biosynthesis. However, the *ilvC* deletion mutant (Δ*PX2605*) grew well in the absence of L-leucine and did not grow in the absence of isoleucine and valine (Figure 6), indicating that other pathways are involved in leucine synthesis. Valine is the putative precursor of leucine, and the first reaction is catalyzed by the aspartate aminotransferase PXO_02420 (AspC). We thus tested the roles of AspC in leucine synthesis and found that Δ*aspC* showed significantly reduced growth in M4N medium in the absence of leucine but grew well in the absence of isoleucine or valine (Figure 6), suggesting that AspC is, in fact, involved in leucine synthesis, although there are other pathways for leucine synthesis because removing of all three types of BCAAs from media did not kill the Δ*aspC* mutant. All of these possible pathways might allow good growth of *Xoo* under the leucine-free condition.

Interestingly, the growth rates of Δ*aspC* were lower than those of wild-type PXO99A under all of the tested conditions (Figure 6). The *AspC* mutation led to the generation of small cells with fewer origins and slow growth in *E. coli*, demonstrating that AspC-mediated aspartate metabolism coordinates the *E. coli* cell cycle [23]. The sequence conservation of AspC between *E. coli* and *Xoo* and the slow growth phenotypes of Δ*aspC* (Figure 6) suggest that AspC might also have a specific effect in the cell cycle in *Xoo*, which requires further studies to support.

### 4.3. Possible Mechanism of Leucine Regulation of Xoo Pathogenicity

Leucine has diverse regulatory functions in humans, fungi, and bacteria [9,20,24]. For example, leucine regulates iron metabolism in *C. neoformans* [9] and type 1 fimbriae production in *Salmonella enterica* [20]. Our transcriptional analysis, however, showed that a lack of leucine inhibited the expression of chemotaxis and type VI secretion as well as genes of the biosynthesis pathways of some other amino acids (Figure 5). Moreover, Lrp was shown to regulate diverse functions, including pathogenicity, in a leucine-dependent manner in different bacteria [20,21], but our data showed that it could not influence *Xoo* virulence and growth (Figure 1 and Figure 2). These results imply that leucine may regulate these virulence-related pathways in an Lrp-independent manner, which is different from other organisms.

## 5. Conclusions

Our findings highlight that amino acid metabolic pathways are important for *Xoo* growth in planta and that a special amino acid synthesis can be achieved by different pathways, which enables bacteria to survive in amino acid-limited environments. Further studies should focus on the mechanisms and pathways of amino acids in regulating the crosstalk among amino acid metabolism pathways and *Xoo*-host interactions, which may provide new insight into the identification of novel drug targets.

## Figures and Tables

**Figure 1 microorganisms-07-00693-f001:**
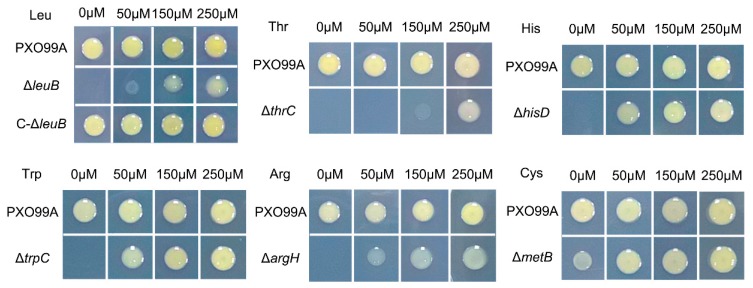
Growth of *Xoo* amino acid auxotrophic strains. Bacterial cultures were spotted and cultured for three days at 28 °C on M4N agar plates that were supplemented with 0, 50, 150, and 250 µM amino acids as indicated in the figures. PXO99A, wild-type *Xoo* strain; Δ*leuB*, *leuB* deletion mutant; Δ*thrC*, *thrC* deletion mutant; Δ*hisD*, *hisD* deletion mutant; Δ*trpC*, *trpC* deletion mutant; Δ*argH*, *argH* deletion mutant; and Δ*metB*, *metB* deletion mutant. Three replicates for each treatment were performed; the experiments were repeated three times.

**Figure 2 microorganisms-07-00693-f002:**
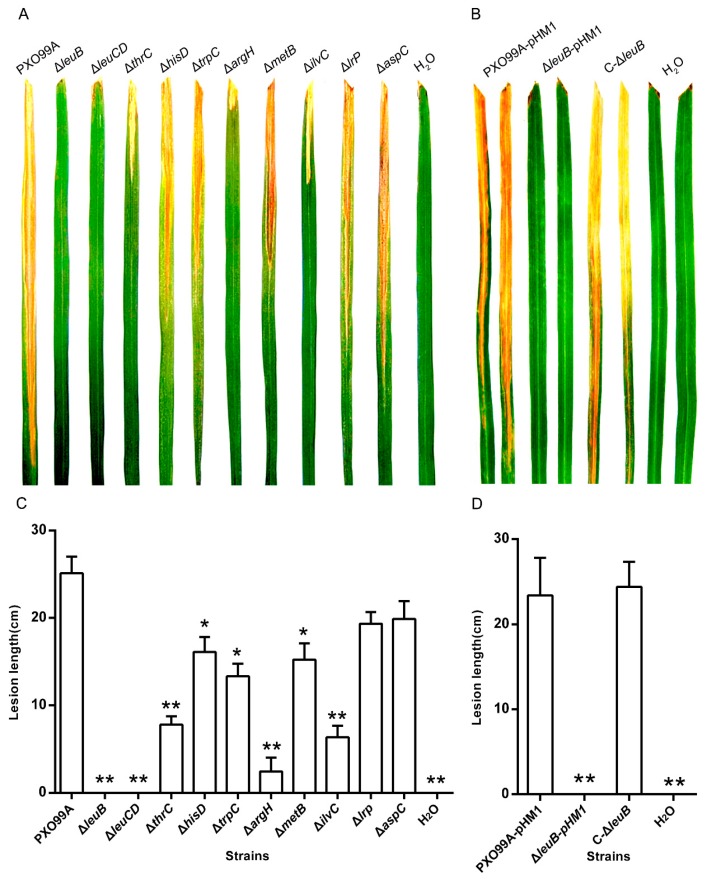
Virulence of *Xoo* strains on rice leaves. Water-soaked lesions displayed on IR24 60-day-old rice leaves with the indicated *Xoo* strains by clipping methods are shown in (**A**,**B**). The lesion length of the infected leaves is shown in (**C**,**D**). Virulence assays were performed on 20 leaves, and the means ± standard deviation were calculated at 14 days post-inoculation. PXO99A, wild-type *Xoo* strain; Δ*leuB*, *leuB* deletion mutant; Δ*leuCD*, *leuCD* deletion mutant; Δ*thrC*, *thrC* deletion mutant; Δ*hisD*, *hisD* deletion mutant; Δ*trpC*, *trpC* deletion mutant; Δ*argH*, *argH* deletion mutant; Δ*metB*, *metB* deletion mutant; Δ*ilvC*, *ilvC* deletion mutant; Δ*lrp*, *lrp* deletion mutant; Δ*aspC*, *aspC* deletion mutant; PXO99A-pHM1, PXO99A containing an empty vector pHM1; Δ*leuB*-pHM1, Δ*leuB* containing an empty vector pHM1; and C-Δ*leuB*, the Δ*leuB* complemented strain. The experiments were repeated three times. *: differences between the mutant and PXO99A (*p* < 0.05); **: significant differences between the mutant and PXO99A (*p* < 0.01).

**Figure 3 microorganisms-07-00693-f003:**
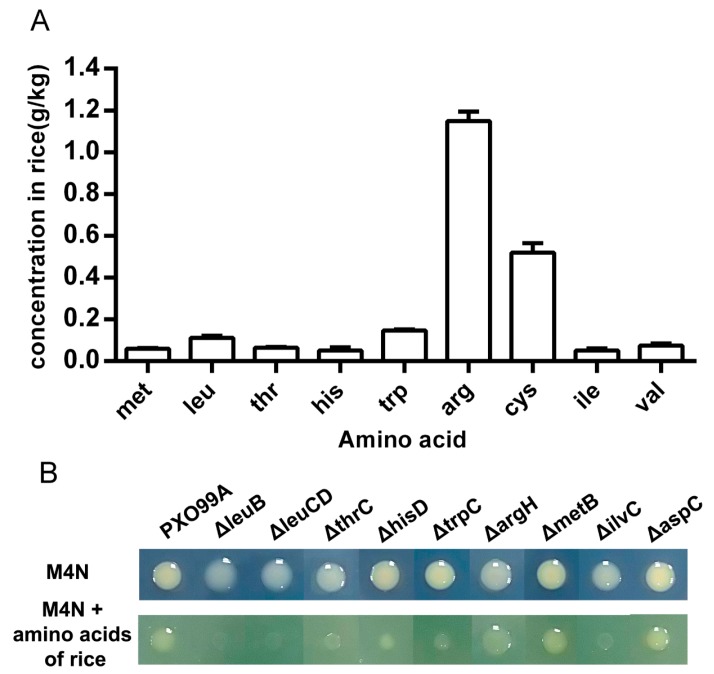
Growth of *Xoo* strains in the presence of amino acids from rice leaves. (**A**) Concentrations of amino acids in rice leaves. (**B**) Growth of *Xoo* strains in M4N minimal medium with supplementation of free amino acids extracted from rice leaves. Bacterial cultures were spotted and cultured for three days at 28 °C on M4N agar plates supplemented with 250 µM amino acids or raw amino acid extracts from rice leaves.

**Figure 4 microorganisms-07-00693-f004:**
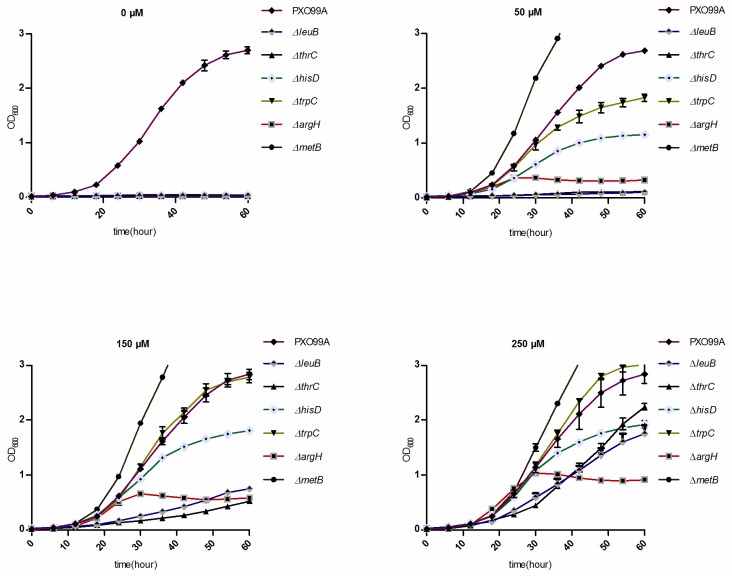
Growth rates of the wild-type and mutant strains under different concentrations of amino acids. The OD_600_ value was determined every 6 h until bacterial growth reached the stationary stage. Three replicates for each treatment were performed; the experiments were repeated three times. Vertical bars represent standard errors. Leucine, threonine, histidine, tryptophan, arginine, and cysteine (concentrations of each are shown on each panel) were added to M4N minimal medium to culture the Δ*leuB*, Δ*thrC*, Δ*hisD*, Δ*trpC*, Δ*argH*, and Δ*metB* mutants, respectively. No amino acids were supplemented in M4N minimal medium during culturing of PXO99A.

**Figure 5 microorganisms-07-00693-f005:**
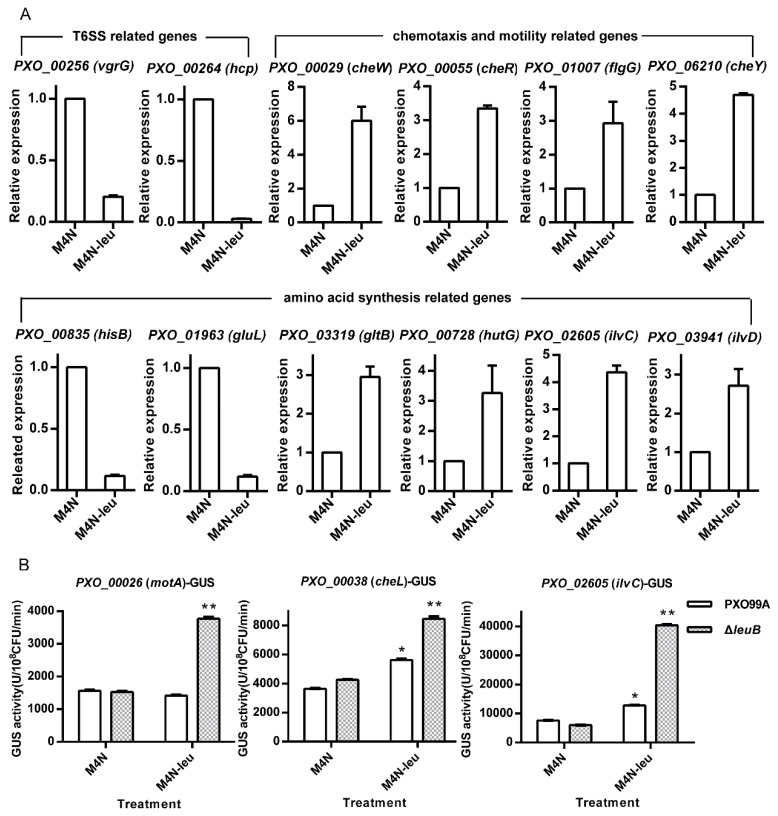
Leucine regulates the expression of genes involved in the biosynthesis of other amino acids and virulence-related genes. (**A**) The relative expression levels of genes involved in chemotaxis, type VI secretion, and amino acid synthesis in Δ*leuB* in M4N medium supplemented with 250 µM leucine or not. (**B**) β-Glucuronidase (GUS) analysis of the activities of chemotaxis- and amino acid synthesis-related gene promoters in Δ*leuB* and PXO99A. The experiments were repeated three times. The data are the means ± standard deviations from three repeats. *: differences between the mutant and PXO99A (*p* < 0.05); **: significant differences between the mutant and PXO99A (*p* < 0.01).

**Figure 6 microorganisms-07-00693-f006:**
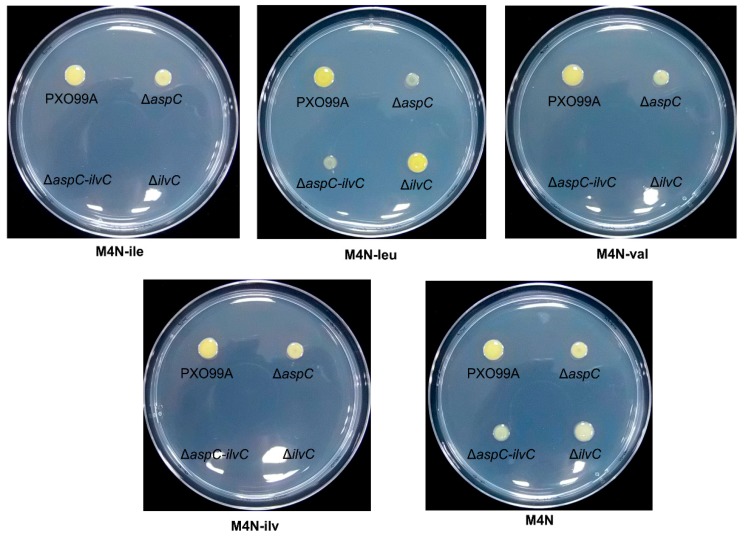
Growth of *Xoo* branched-chain amino acids (BCAA) synthesis mutants. Bacterial cultures were spotted and cultured for three days at 28 °C on M4N agar plates without supplementation with isoleucine, leucine, or valine. Three replicates for each treatment were performed, and the experiment was repeated three times.

**Table 1 microorganisms-07-00693-t001:** Amino acid synthesis-related genes discussed in this study.

Entry	Gene Name	Definition	Related Amino Acids
*PXO_02610*	*leuB*	3-isopropylmalate dehydrogenase	Leu
*PXO_02612*	*leuD*	3-isopropylmalate dehydratase, small subunit	Leu
*PXO_02613*	*leuC*	3-isopropylmalate dehydratase, large subunit	Leu
*PXO_00822*	*thrC*	threonine synthase	Thr
*PXO_00833*	*hisD*	histidinol dehydrogenase	His
*PXO_04004*	*trpC*	indole-3-glycerol phosphate synthase	Trp
*PXO_00346*	*argH*	argininosuccinate lyase	Arg
*PXO_03157*	*metB*	cystathionine gamma-synthase	Cys
*PXO_03072*	*lrp*	leucine responsive regulatory protein	
*PXO_02605*	*ilvC*	ketol-acid reductoisomerase	Ile, Val
*PXO_02420*	*aspC*	aspartate aminotransferase	Leu, Ala, Arg, et al.

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
