# Peer review of "Biosynthesis of Amino Acids in Xanthomonas oryzae pv. oryzae Is Essential to Its Pathogenicity"

_microorganisms, 2019, doi:10.3390/microorganisms7120693_

Round 1

Reviewer 1 Report

Letter to Authors,

This manuscript shows that some amino acids influence Xoo growth and virulence. This fact is reasonable and important. Furthermore, the authors conduct research and suggest that Leucine regulates gene expression patterns of chemotaxis and type VI secretion. This study will be a significant contribution to our knowledge of the growth and virulence of Xoo.

Overall comments:

This is a well-written manuscript. The results are convincing, and the conclusions are supported by the data. Especially, the research planning and figures were well designed.

Major comments:

Check the figure numbers in discussion section (e.g. in L. 257 and 258, Fig. 3 -> Fig. 2. In L. 260, Fig. 4 -> Fig. 3. In L. 261, Fig. 3 and 4 -> Fig. 2 and 3). These mistakes are highly confusing.

Minor comments:

53. Describe not for abbreviation “Branched-chain amino acids (BCAA)”. 82. Describe methods not company name “Qingdao Aci tech ...”. 180 and 183. Insert (Fig. 3B) into the end of the sentences. 197 and 198. (ΔPX0822, ΔPX0833, ΔPX4004, ΔPX0346 and ΔPX3157) -> (ΔthrC, ...)

Table S1. In the line of pK18mobsacB, describe KmR instead of KanR.

Fig. S3. What does mean the three PCR products? Biological replicate or technical replicate?

Reviewer 2 Report

The introduction need more information about the role of amino acids in pathogenicity.  gene deleted for amino acid synthesis may be listed in the abstract. In Table S1 X00 no genotype information listed. Is there any other AA studied along with leucine. The current research highlight that amino acids metabolic pathways are important for Xoo growth in planta but maybe there are several other also play role in in the pathogenicity. Along with that the Plant genotype and the defense mechanism also responsible for the pathogenicity. 
It is very difficult to make concrete comments about the role of a single amino acid in the pathogenesis. 

Reviewer 3 Report

Suggested revisions:

Major: Extensive language editing required

        Detail discussion and conclusion required for clarity

Details:

P2L73 What is PXO_2610: explain

P3L108 Which amino acids, all or selected: explain

P3L116: Because we can not find…: restructure the sentence

P3L122: And ..whether these genes are special to…: whether these genes are essential/required for..

P3L123: of corresponding amino acid: of the corresponding..

P3L126: had significant: showed significant

P3L134: can not: did not

P6L186: free amino acids in rice leaves may restrict Xoo growth in planta: limiting amino acid availability in rice leaves may restrict Xoo growth in planta:

P6L193: 3.4. The Usage Abilities Are Different among Amino Acids by Xoo: restructure the section title, does not make sense

P6L196: Usage abilities…restructure the sentence

Figure 4 legend is not self-explanatory and does not match the text. Does the graph labelled as 50 um represents all amino acids present in medium at 50 um concentration each?

Maintain consistency in active and passive voices and past/present tense throughout the manuscript.

P9L256: requires lots of molecules to sustain its growth in planta: use scientific language/lots of molecules -needs appropriate reference in context of amino acid requirement for Xoo growth

P9L259: use of tense

P9261: amino acids contents: either acids or contents

The role of AspC: AspC-Mediated aspartate Metabolism is known to coordinate the cell cycle in E.Coli. A more in-depth discussion would have been supportive of the conclusion the authors have made.

A more clear-cut conclusion and discussion should be included with appropriate diagrams to pinpoint the possible role of leucine and other amino acids in the growth process of Xoo
